# A Rotating Object Detector with Convolutional Dynamic Adaptive Matching

Leibo Yu [1], Yu Zhou [2], Xianglong Li [1], Shiquan Hu [1] and Dongling Jing [3,*]

1   Computer Science and Technology, Wuhan University of Science and Technology, Wuhan 430065, China
2   Wuhan Maritime Communication Research Institute, Wuhan 430200, China
3   Information Engineering, Beijing Institute of Technology, Beijing 100081, China
*   Correspondence: jingdonglin@bit.edu.cn

**Abstract:** Standard convolution sliding along a fixed direction in common convolutional neural networks (CNNs) is inconsistent with the direction of aerial targets, making it difficult to effectively extract features with high-aspect-ratio and arbitrary directional targets. To this end, We have fully considered the dynamic adaptability of remote sensing (RS) detectors in feature extraction and the balance of sample gradients during training and designed a plug-and-play dynamic rotation convolution with an adaptive alignment function. Specifically, we design dynamic convolutions in the backbone network that can be closely coupled with the spatial features of aerial targets. We design a network that can capture the rotation angle of aerial targets and dynamically adjust the spatial sampling position of the convolution to reduce the difference between the convolution and the target in directional space. In order to improve the stability of the network, a gradient adaptive equalization loss function is designed during training. The loss function we designed strengthens the gradient of high-quality samples, dynamically balancing the gradients of samples of different qualities to achieve stable training of the network. Sufficient experiments were conducted on the DOTA, HRSC-2016, and UCAS-AOD datasets to demonstrate the effectiveness of the proposed method and to achieve an effective balance between complexity and accuracy.

**Keywords:** remote sensing detection; convolutional neural network; dynamic network; rotational convolution





## 1. Introduction

Aerial object detection is used to automatically identify objects on surfaces or in air, such as buildings, roads, vehicles, etc. Its main purpose is to analyze aerial image data and to use them for resource management, environmental monitoring, urban planning, military reconnaissance, disaster management, etc. With the development of deep learning and GPU performance, the application of convolutional neural networks in RS detection is becoming increasingly widespread.

The object detection framework is usually divided into one-stage [1–3] and two-stage [4–6] detectors. A one-stage detector utilizes a horizontal anchor box to generate candidate regions and then performs regression and classification operations to pinpoint the target accurately. SPP-Net [7] uses a spatial pyramid pooling strategy to generate fixed-size features without being affected by image scale. SSD [8] achieves end-to-end object detection by simultaneously predicting the position and category of targets on different levels of feature maps. However, these methods can lead to misalignment and background interference between the target and candidate regions. In order to solve the problem, a two-stage detector achieves alignment between the target and the candidate region through multiple fine refinements of the candidate regions. All candidate boxes generated by RPN (including positive samples and the background) would perform nonmaximum suppression to remove highly repetitive boxes, resulting in the final set of proposal boxes. The Roi transformer [2] improves the accuracy of target localization by finely adjusting

the internal features of candidate boxes, thereby eliminating background interference. Yang [1] designed a new multi-category rotation detector, SCRNet, for small, dense, and rotating objects, which integrates multi-layer features into anchor sampling to improve sensitivity for detecting small targets. However, such techniques often necessitate intricate calculations to eliminate redundant and overlapping candidate boxes, posing challenges in portability across distinct detectors and leading to increased inference times, rendering them unsuitable for embedded platforms with limited computational capabilities.

The parameters and structure of these static networks mentioned above are fixed during inference and are not suitable for aerial targets with high aspect ratios and arbitrary directional orientations. At present, research on RS object detection has undergone a transformation from static networks to dynamic networks. Specifically, the structure or parameters of dynamic networks can adapt to different inputs and have significant advantages in accuracy, computational efficiency, and adaptability. CAT [9] is a recursive neural network model that adaptively calculates time, allowing the model to dynamically adjust the calculation time when processing sequence data to more effectively adapt to inputs of different step sizes. However, in some cases, the generalization ability of this method may be limited for different tasks, especially on test data with significant differences from the train data. Shazeer [10] proposed a neural network structure called the Sparsely-Gated Mixture-of-Experts layer (MoE), which can support a large model scale and achieve efficient training and inference through effective gating mechanisms. However, the model structure is relatively complex, occupying a large amount of storage space, and the training and inference time would increase.

In summary, the previous dynamic network had high computational complexity, making it difficult to deploy and apply to various detectors. Specifically, aerial targets such as planes and ships often have a high aspect ratio and are arranged in multiple directions. Existing standard convolution structures slide horizontally, which can lead to angular differences with aerial targets. In samples with high aspect ratios, the majority of preset boxes that meet the IoU threshold fail to fully encompass the target, and the angle between these preset boxes and the ground truth boxes is significant. The angle difference between low-quality samples and truth labels is significant, resulting in significant gradients during the training process. However, high-quality samples with high cover and ground truth boxes have smaller gradients, which contribute less to training losses. In general, low-quality samples with a higher proportion contribute more to the loss, while high-quality samples with a lower proportion contribute less to the loss, resulting in unstable training.

We design an adaptive rotating target detector with sample dynamic adjustment, which achieves plug-and-play and high-precision detection without increasing computational complexity. In the backbone network, an angle estimator that matches the target orientation is constructed. The spatial sampling position of the convolution kernel is rotated, and a plug-and-play dynamic convolution layer is designed to ensure that the convolution operator for feature extraction aligns with the target in the spatial direction. The loss function we designed amplifies the gradient of samples with high IoU values and low regression gradients, improving the contribution of high-quality samples to regression loss. By dynamically adjusting the gradients generated by the samples during the training, we ensure efficient and stable training of the model. Our contributions are as follows:

(1) We systematically analyze the problem of feature misalignment and imbalanced gradients caused by different quality samples during the training process of the current RS detector.
(2) In the feature extraction stage, we design a plug-and-play rotating dynamic convolution that can adaptively align the convolution with the target direction based on the spatial distribution of the target.
(3) In training, we design a gradient adaptive equalization loss function to optimize the contribution of gradients from different samples to regression loss and improve the training stability.

The organizational structure is as follows: In Section 2, rotating object detection and dynamic networks are introduced. Section 3 proposes two innovative designs, namely dynamic convolution aligned with the target direction and gradient adaptive equalization loss function for sample balancing. Section 4 conducted a series of comparison and ablation experiments on three public datasets to demonstrate the effectiveness of our method. Section 5 summarizes and analyzes the proposed methods.

## 2. Related Work

In this section, we first review the main work on rotating object detection and then introduce relevant research on dynamic networks. It is worth noting that most of these tasks require large-scale adjustments to the network structure, making it difficult to apply.

A.   Rotating object detection

The detector [11] based on rotating boxes effectively solves the problem of severe background interference and repeated detection boxes in the field of aerial targets. For rotating target detectors, the current research mainly focuses on three aspects: rotation feature extraction, rotation box generation, and loss function optimization.

**Rotation feature extraction**: at present, most rotation target detectors use ResNet as the backbone network to extract the angle feature through the design of the improved convolution module. Azimi [12] designed a cascaded network that generates rotational features in the backbone network through feature pyramid networks and multi-scale convolution kernels. This makes target direction detection more robust, but it can lead to a complex model structure. Ran [13] designed a lightweight rotation detector and added an enhanced channel attention (ECA) module in each layer to strengthen the representation ability of the model and to improve the detection performance. However, the features of small targets are easily lost during downsampling. RODFormer [14] uses a structured transformer to collect features of different resolutions, which is convenient for detecting targets densely distributed from multiple angles in aerial images.

**Rotation box generation**: Ding [2] designed a rotation region of interest learner (RRoI Learner) module that adds a small amount of computation to apply spatial transformation to RoIs and generates rotation boxes to solve the problem of misalignment between horizontal boxes and targets. However, this method ignores angle deviation and is prone to missing targets with large aspect ratios. Xu [3] generated a rotation box by sliding four vertices on the horizontal box. This rotation box generation method has a small angle deviation, but it also has the problem of inaccurate boundary box regression and slow inference speed. When the object is in a 3D environment, the quaternion [15] can be used to rotate the target. The quaternion cannot be applied to 2D environments, so we use matrices for rotation transformation.

**Regression loss function optimization**: GWD [16] uses the Gaussian Wasserstein distance instead of the nondifferentiable rotation IoU to optimize the regression loss function, align model training with measurement standards, and solve the problem of a discontinuous rotation angle range. Yang [17] proposed an approximate SkewIoU loss based on the Gaussian product, which solves the problem of gradient explosion or vanishment during the training of rotation detectors.

However, the designed models are often too complex and difficult to transplant to different detectors. The variable rotation convolution kernel proposed in this paper serves as a plugin module, which is more convenient to embed into the detector and improves the model's detection performance for rotating targets.

B.   Dynamic network

A network that adjusts its structure or parameters based on different models during inference is called dynamic network [18,19], which is an emerging topic in deep learning. Compared to static network, a dynamic network improves the inference efficiency and compatibility of the model. From the perspective of data processing and training parameter

optimization, the dynamic network is mainly divided into two categories: spatial dynamic networks and sample dynamic networks.

**Spatial dynamic networks** focus on areas with the highest correlation with the target and perform spatial adaptive inference on these areas, while areas with little correlation with the target can be ignored. Yang [20] proposed a resolution adaptive network (RANet), which contains multiple deep subnets with different weights. The samples are first identified from the subnet with the smallest weight. If the results meet the conditions, the sample exits the network early. Otherwise, the samples are input to a subnet with higher weights for identification, achieving a balance between accuracy and computational complexity and reducing spatial redundancy. Ming [21] proposed a sparse label assignment strategy (SLA) that selects high-quality sparse anchors based on the fixed IoU, dynamically balancing the inconsistency of classification and regression during training.

**Sample dynamic networks** can be subdivided into dynamic architecture and dynamic parameters based on different data processing methods. The dynamic architecture dynamically adjusts the network structure based on different input samples, effectively reducing redundant calculations. BranchyNet [22] has an additional branch classifier that allows test samples to exit the network early through these classifiers. SkipNet [23] is a residual network with gating units that selectively skip unnecessary convolution layers during inference, significantly reducing the inference time of the model. When the input data are fixed, the network parameters can be adjusted to improve the feature extraction effect, which will lead to a small increase in computational costs. Su [24] proposed pixel-adaptive convolution (PAC), which multiplies filter weights with convolution kernels with spatial variations to address the limitations of the content adaptation layer. The proposed method with dynamic parameter specificity improves the feature extraction ability of the backbone by transforming the sampling positions of the convolutional kernel. The detection effect on rotating targets is particularly significant.

## 3. Method

Unlike previous work, we systematically analyzed the impact of multi-directional targets on RS detectors during feature extraction and training. Instead of devising a complex network architecture, it is imperative to craft an adaptive feature extractor that aligns and rotates effectively. Then, the rotation direction of the convolution kernel is dynamically adjusted to adaptively align with the target orientation. The gradients of the model are further simultaneously optimized during training to avoid gradient imbalance caused by large gradient sample data. The overall structure is shown in Figure 1. In particular, the features extracted from the backbone network are input into the dynamic convolution module (DCM), where efficient aggregation of features is achieved through convolutional operations and average pooling. Two activation function branches were designed in DCM to predict the angle and weight of convolution rotation. The spatial sampling of the convolution kernel's rotation direction is determined by both rotation angle and weight, utilizing a rotated convolution kernel for multi-directional feature extraction. Then, the feature pyramid network constructs classifiers and regressors based on the multi-directional features. We design a gradient adaptive equalization loss function during training to balance samples with different gradient contributions and to improve training efficiency.

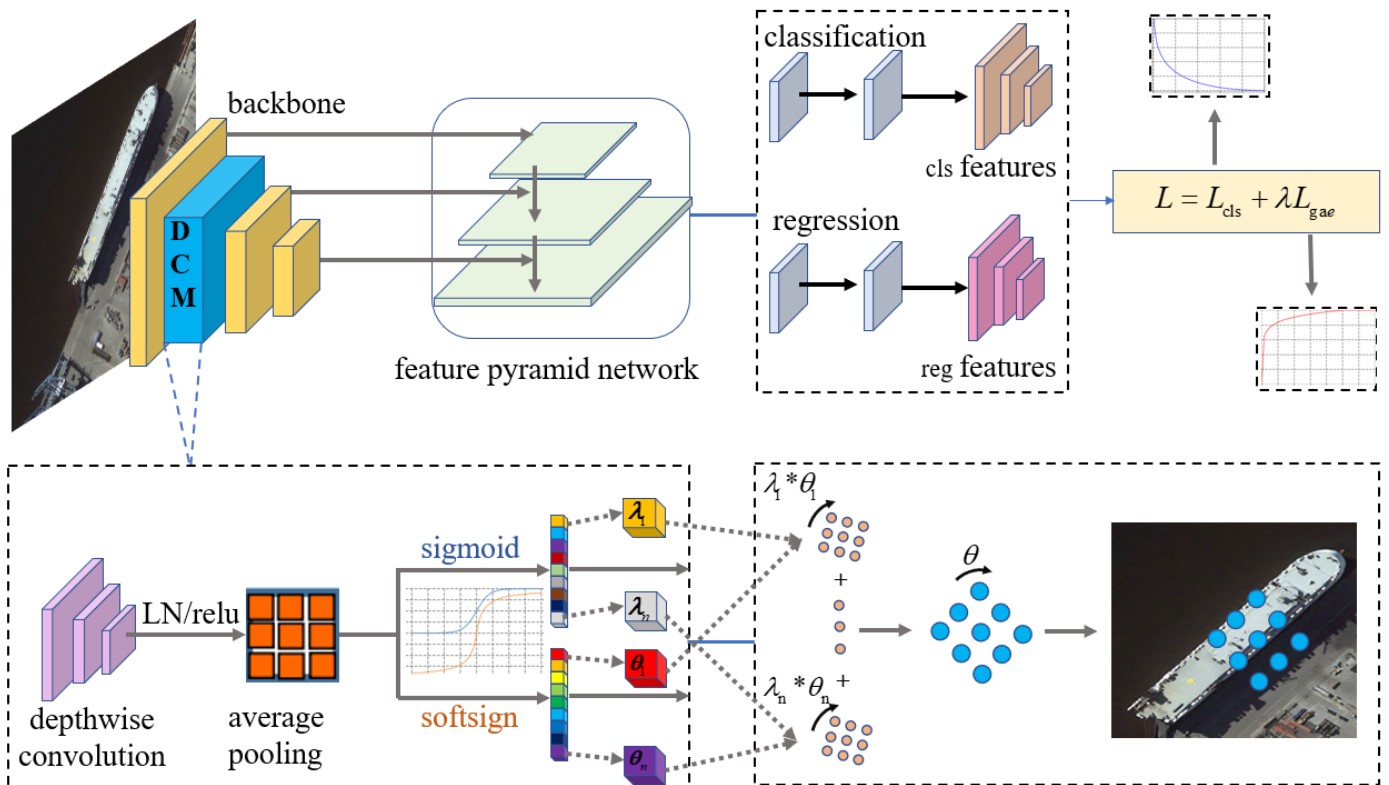

**Figure 1.** The framework we propose consists of three parts: Dynamic Convolutional Module (DCM), Feature Pyramid, and Classification Detection Head. Firstly, the target angle is predicted using DCM and the convolutional kernel is rotated. Then, the extracted rotational features are input into the feature set pyramid to fuse multi-scale information. Finally, a gradient adaptive equalization loss function is used to perform equalization on the network.

### 3.1. Dynamic Convolution

As shown in Figure 2, we rotate the convolution to make the network more suitable for arbitrary directional RS object detection.

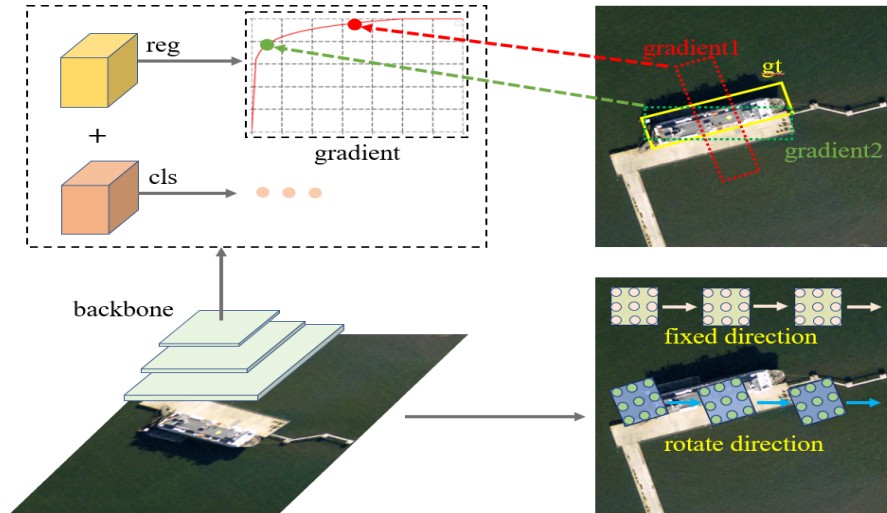

**Figure 2.** Firstly, to address the issue of feature errors in RS detection, we dynamically rotated the convolution to effectively extract features from multi-directional targets. Secondly, in order to solve the problem of imbalanced gradients in high-quality positive samples, a gradient adaptive equalization loss function was designed.

**Spatial angle prediction**: A feature with the size $[C_{in}, H, W]$ is input into the DCM module, firstly, through the calculation of $3 \times 3$ convolution kernels $W_{3 \times 3}$, followed by ReLU activation $relu(\cdot)$ and average pooling $ave(\cdot)$ to achieve efficient aggregation of the feature. The formula is expressed as follows:

$$\mathbf{x}' = ave(relu(W_{3 \times 3} * \mathbf{x})) \tag{1}$$

Then, the feature is input into two branches separately. The first branch consists of a linear layer and a softsign activation function. The range interval of the softsign activation function is $[-1, 1]$, and the curve is smooth, used to predict the rotation angle $\theta = [\theta_1, \theta_2, \cdots, \theta_n]$. The linear layer deviation is set to False to avoid angle errors. The formula is as follows:

$$\theta = softsign(\mathbf{x}') \tag{2}$$

where $\theta$ represents the angle of the predicted rotating target, and $\mathbf{x}'$ is the feature of the rotating target extracted from the backbone network. The purpose of Formula (2) is to output the rotated target features extracted by the network as parameters to the softsign activation function and to calculate the angle of the rotated target. This angle serves as a parameter for rotational convolution, ultimately aligning the convolution with the remote sensing target in the spatial direction. $softsign(\cdot)$ can be expressed as follows:

$$softsign(\mathbf{x}) = \frac{\mathbf{x}}{1 + |\mathbf{x}|} \tag{3}$$

The second branch consists of a linear layer with bias and a sigmoid activation function. The range interval of the sigmoid activation function is $[0, 1]$, which is used to predict the combination weight $\lambda = [\lambda_1, \lambda_2, \cdots, \lambda_n]$, and the formula is as follows:

$$\lambda = sigmoid(\mathbf{x}') \tag{4}$$

where $\lambda$ is a set of weight vectors corresponding to $\theta$. By weighted summation, the final prediction of the rotation target angle is made. $sigmoid(\cdot)$ can be expressed as follows:

$$sigmoid(\mathbf{x}) = \frac{1}{1 + exp(-\mathbf{x})} \tag{5}$$

Due to the possibility of multiple targets being in the same image, considering spatial richness, spatial angle prediction is used to calculate a set of angles and a set of weights and to match the best rotational convolution kernel for different targets through weighted summation. The rotation angle $\theta$ and multi-angle weighting factor $\lambda$ obtained in the spatial angle prediction section are used as parameters and input into the second part of DCM to generate a rotation convolution kernel.

**Dynamic rotation convolution**: If standard convolution is used for the feature extraction of multi-directional targets, the features shown in Figure 3a would be obtained. It can be seen that there is an angle between the convolution and the target, which can lead to an offset between the extracted features and the target, resulting in a large amount of invalid information in the features.

DCM has n convolutions kernels $\mathbf{W} = (W_1, W_2, \cdots, W_n)$, each with a shape of $[C_{in}, C_{out}, k, k]$, and $C_{in}$ and $C_{out}$ representing the number of input and output channels. The size of the convolution kernel is $k \times k$. Firstly, a single rotational convolution kernel $W'$ is calculated, taking $\theta_i$ and $W_i$ as parameters and using the $Rotate(\cdot)$ function:

$$W'_i = \text{Rotate}(W_i, \theta_i), i = 1, 2, \ldots, n \tag{6}$$

As shown in Figure 3b, each convolution kernel $W_i$ is rotated clockwise by an angle of $\theta_i$, and each convolution kernel is added according to the weight $\lambda_i$ to obtain a spatially aligned rotated convolution kernel $W'$ with the target. Then, the rotational convolution kernel is used to extract features from the target, as shown in Figure 3c. By comparing the feature extraction methods of standard convolution and rotational convolution, it is

evident that our approach accounts for the directional characteristics of the target, making it more suitable for the extraction of arbitrary directional aerial target features.

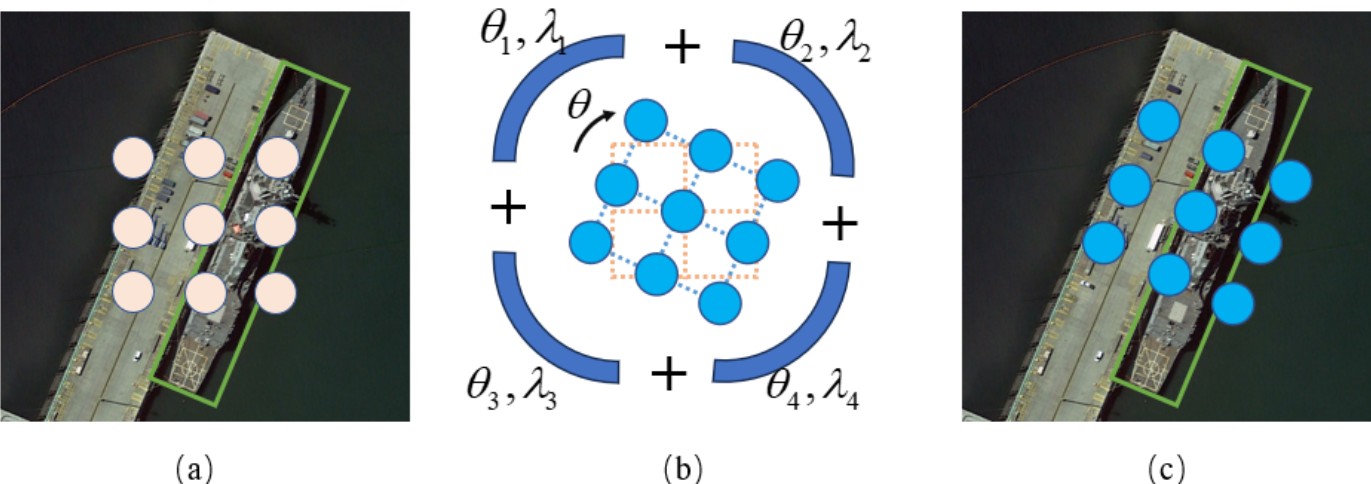

**Figure 3.** (**a**) Standard convolution for feature extraction (**b**) DCM predicts the rotation angle of convolution (**c**) Rotational convolution for feature extraction. DCM Predicts multiple angle branches with different weights. Weighted sum of these angle branches to predict the rotation angle of multi-directional remote sensing targets.

The rotation convolution kernel $W'$ and feature perform convolution operations to extract rotation features. In order to establish the correlation between feature channels, the rotational features of each part are summed according to their weights to calculate the aligned rotational feature with the target:

$$y = \lambda_1 \left(W_1' * x\right) + \lambda_2 \left(W_2' * x\right) + \cdots + \lambda_n \left(W_n' * x\right) \tag{7}$$

Formula (7) has multiple convolution operations. According to the distribution law of convolution, it can be written as follows:

$$y = \left(\lambda_1 W_1' + \lambda_2 W_2' + \cdots + \lambda_n W_n'\right) * x \tag{8}$$

Formula (8) indicates that the rotation weight of the combination is first calculated, and then, convolution operation is performed with feature $x$. Compared to the standard convolution, the newly added computational complexity of the convolution operation can be ignored. Moreover, through dynamic rotation convolution, the network is more sensitive to the angle features of aerial targets.

*3.2. Gradient Adaptive Equalization Loss*

Aerial targets often have the characteristics of a high aspect ratio, multi-directionality, and a dense layout. Most mainstream networks have horizontal detection boxes. If directly applied to aerial targets, the detection boxes will contain a large amount of background information, and invalid background information would lead to a decrease in detection accuracy. These mainstream networks adopt a multi-task learning mode, which trains both classification and regression tasks simultaneously. The loss function is defined as follows:

$$L = L_{cls}(k, u) + \lambda L_{reg}(p, v) \tag{9}$$

where $l_{cls}$ and $l_{reg}$ represent the loss functions for classification and regression, and $k$ and $p$ represent the corresponding predicted values. $u$ and $v$ are the corresponding labels. $\lambda$ is a hyperparameter that adjusts the regression loss weight in multi-task learning. The regression loss function usually uses a smooth L1, defined as follows:

$$L_{smooth}(x) = \begin{cases} 0.5x^2 & \text{if } |x| < 1 \\ |x| - 0.5 & \text{otherwise} \end{cases} \tag{10}$$

The partial derivative of Formula (10) is calculated to obtain the gradient function of the smooth L1:

$$\frac{\partial L_{smooth}}{\partial x} = \begin{cases} x & \text{if } |x| < 1 \\ \pm 1 & \text{otherwise} \end{cases} \tag{11}$$

RS object detection focuses on the accuracy of the regression box. If the weight of the $\lambda$ is directly increased, it will make the abnormal samples with large gradients and severe background interference in the regression loss more sensitive, which is unfavorable for training. From Formula (11), it can be seen that for the parts with larger gradients, the calculated losses are also larger, which leads to the model paying more attention to abnormal samples. In comparison to abnormal samples, some high-quality samples in aerial targets possess smaller gradients during training, leading to insufficient contributions to the losses. The smooth-L1 loss function did not adequately train this portion of high-quality samples. As shown in Figure 2, a gradient adaptive equalization loss function was designed to improve the stability of the RS detection network.

Therefore, we design the gradient adaptive equalization loss function $L_{gae}$ and first provide the definition of its gradient function:

$$\frac{\partial L_{gae}}{\partial x} = \begin{cases} a \ln(b|x| + 1) & \text{if } |x| < 1 \\ c & \text{otherwise} \end{cases} \tag{12}$$

If the upper limit of the gradient is not set, training will pay more attention to some abnormal samples with large gradients, while ignoring high-quality samples, which will cause uneven training. Therefore, the upper limit of the gradient $c$ is set. In order to ensure the continuity of the gradient, the relationship between the equilibrium parameters $a$, $b$, and $c$ is as follows:

$$a \ln(b+1) = c \tag{13}$$

From Figure 4a, it can be seen that compared to the smooth L1 gradient function, $L_{gae}$ amplifies the value of the small gradient, indicating that the network pays more attention to the regression loss of high-quality samples during training. Moreover, as the equilibrium parameter $a$ increases, the small gradient value also increases without affecting the large gradient value part. Finally, the weight of the high-quality samples in training are increased, the gradient function is integrated, and the loss function $L_{gae}$ is obtained:

$$L_{gae}(x) = \begin{cases} \frac{a}{b}(b|x| + 1) \ln(b|x| + 1) - a|x| & \text{if } |x| < 1 \\ c|x| & \text{otherwise} \end{cases} \tag{14}$$

In this experiment, $a = 0.5$ and $c = 1.5$.

$L_{gae}$ performs gradient amplification on high-quality samples, thereby increasing the contribution of high-quality samples to regression losses and promoting network equalization training.

To ensure the balance of sample categories, the classification loss function Focal Loss is selected, defined as follows:

$$L_{cls}(k, u) = \begin{cases} -\log(k) & \text{if } u = 1 \\ -\log(1 - k) & \text{otherwise} \end{cases} \tag{15}$$

The overall loss function is as follows:

$$L = L_{cls}(k, u) + \lambda L_{gae}(x) \tag{16}$$

Among them, hyperparameter $\lambda = 0.5$.

In this way, DCM performs dynamic rotation operations on convolution, improving the fitting ability between convolution and the target and enhancing the model's ability to extract directional features. A gradient adaptive equalization loss function is proposed to balance the loss gradient of high-quality samples during training, making model training more efficient.

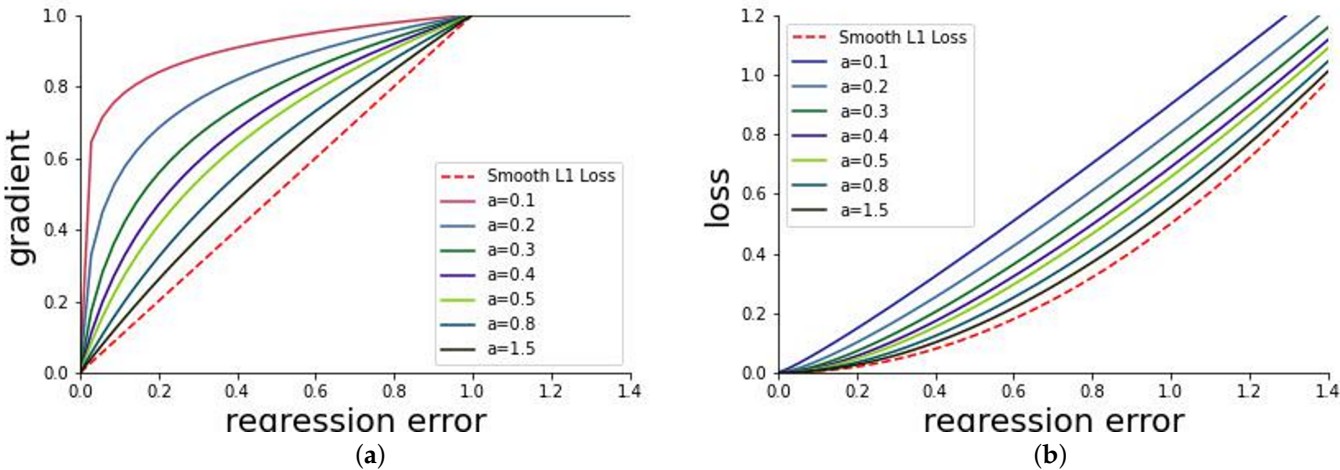

**Figure 4.** (**a**) The relationship between regression values and gradients. (**b**) The relationship between regression values and losses; the smaller the regression value, the smaller the difference between the predicted results of the model and the true labels, which usually occurs on high-quality samples.

## 4. Experiment

### 4.1. Datasets

HRSC2016 [25] is a publicly available RS ship dataset, which features densely arranged and diverse directions of docked ships, with complex image backgrounds. The texture of the ship is similar to that of the shore, and there are significant differences in the scale of multiple targets in one image. This dataset consists of 1061 images, with a resolution ranging from $300 \times 300$ to $1500 \times 900$. For the detection accuracy of HRSC2016, we used the average precision (AP) as the evaluation standard, which is consistent with PASCAL VOC 2012.

UCAS-AOD [26] is a aerial image dataset used for detecting two types of targets: cars and airplanes. Among them, the plane dataset has 600 images including 3210 airplanes, and the automotive dataset has 310 images including 2819 vehicles. All images were carefully selected, and the targets are densely arranged with significant directional differences. We randomly divided it into training set, validation set, and testing set, with a distribution ratio of 6:2:2.

DOTA [27] is a public large-scale rotating object detection dataset that includes 2806 aerial and satellite images, as well as 188,282 annotation boxes. The image resolution span is large, ranging from $800 \times 800$ to $4000 \times 4000$. These images mainly come from Google Earth, a dataset with 15 categories, and the targets in the images are densely arranged, with diverse directions and some targets obstructed, reflecting real-world scenes.

### 4.2. Evaluation

In RS object detection tasks, we commonly use precision $P$ to represent the proportion of correctly predicted true-positive samples to the total number of predicted positive samples. The recall rate $R$ represents the proportion of the correctly predicted positive samples to the actual number of positive samples.

$$P = \frac{TP}{TP + FP} \tag{17}$$

$$R = \frac{TP}{TP + FN} \tag{18}$$

Among them, *FP* represents predicting negative class errors as positive class numbers, and *FN* represents predicting positive class errors as negative class numbers. However, accuracy and recall are mutually exclusive metrics, as achieving high levels of one often results in suboptimal performance for the other. In order to comprehensively evaluate the performance of the model, this paper selects mean average precision (mAP) as the measurement standard for RS object detection. The mathematical meaning of average accuracy AP is the area of the P-R curve, as follows:

$$AP = \int_0^1 P(R)dR \tag{19}$$

When the detection object has *J* categories, mAP is defined as follows:

$$mAP = \frac{1}{J}\sum_{i=1}^{J} AP_i \tag{20}$$

The complexity of the algorithm is measured by the number of parameters (Params) and the number of floating-point operations (FLOPs). The Params formula is as follows:

$$Params = C_{out} \times (k_w \times k_h \times C_{in} + 1) \tag{21}$$

Among them, $C_{out}$ represents the number of output channels, $C_{in}$ represents the number of input channels, and $k_w \times k_h$ represents the size of the convolution.

### 4.3. Parameter Settings

For fair comparison with other methods, all experiments were implemented on PyTorch. The environment was Ubuntu 18.04, Python 3.8, and Python 1.7.0. We chose to pre-train the ResNet101 [28] model on ImageNet as the backbone network for RS object detection. All experiments were conducted using NVIDIA Titan X GPU, limited by the number of GPUs. The batch size was set to 8, and the epochs for model training were 400. The initial learning rate was 0.02, and the network was trained by the Adam optimizer. The initial momentum and weight attenuation were set to 0.9 and $5 \times 10^{-4}$, respectively. In order to eliminate the impact of randomness, each group of experiments was conducted three times and the average was taken to obtain accurate experimental data.

### 4.4. Ablation Studies

4.4.1. Evaluate Different Modules

To verify the improvement in dynamic convolution and gradient adaptive equalization loss functions in RS object detection performance, we conducted ablation experiments on the HRSC-2016 and UCAS-ADO datasets. Table 1 lists the experimental results of this model on the HRSC-2016 dataset. The mAP of the baseline model is only 84.35%, which is because ordinary convolutions slide horizontally along the axis, making it difficult to model high-aspect-ratio and multi-directional targets. When DCM modules were added to the backbone network, mAP increased by 4.34%. This indicates that the dynamic rotation convolution proposed in this paper is more sensitive to the direction of aerial targets, improving the network's ability to extract directional features. After using the loss function GAE, mAP improved by 2.88%. This suggests inadequate training of high-quality samples in aerial targets. By optimizing the loss function and enhancing the gradients in this segment, the training performance of the model can be improved. When both DCM and GAE are added to the network model, mAP increases by 5.79% and reaches a maximum value of 90.14% This indicates that the dynamic convolution and gradient adaptive equalization loss functions could enhance the network's ability to model the target spatial direction while maintaining training stability, thereby significantly improving detection performance.

**Table 1.** Effects of different modules on the HRSC-2016.

| With DCM | With GAE | mAP (%) |
|:---:|:---:|:---:|
| ✗ | ✗ | 84.35 |
| ✓ | ✗ | 88.69 |
| ✗ | ✓ | 87.23 |
| ✓ | ✓ | **90.14** |

DCM: the dynamic convolution module; GAE: the gradient adaptive equalization loss function.

We achieved similar experimental results on the UCAS-ADO dataset. Table 2 compares individual modules and proves that the superposition of two modules has the most significant improvement effect on the network. In feature extraction, DCM improves the model's ability to extract direction features. During training, GAE balanced the sample gradient and increased the contribution of high-quality samples to the loss. There is no conflict between the DCM and gradient adaptive equalization loss function, and when used simultaneously, the mAP reaches 90.52%.

**Table 2.** Effects of different modules on the UCAS-AOD.

| With DCM | With GAE | mAP (%) |
|:---:|:---:|:---:|
| ✗ | ✗ | 86.79 |
| ✓ | ✗ | 88.14 |
| ✗ | ✓ | 87.43 |
| ✓ | ✓ | **90.52** |

4.4.2. Evaluate DCM

To further validate the effectiveness of the proposed DCM, Table 3 validated the impact of convolution number n on network performance on the HRSC-2016 dataset. $n = 0$ is the baseline model without the addition of DCM, and as the number of convolution kernels increases, mAP also continuously increases. When $n = 1$, it indicates that only one rotation convolution kernel is added, which increases mAP by 1.14% compared to the baseline model. This indicates that the rotation convolution can better extract the features of rotating targets. When $n = 4$, mAP reaches its maximum value of 90.41%, indicating that using multiple rotation convolution weighted summation is more effective. As the value of n continues to increase, mAP actually decreases. This indicates that when DCM has four rotation angle branches, dynamic rotation convolution has extracted enough directional feature information. The rotational characteristics of the target are limited, and increasing the value of n does not enhance the model's performance. Nevertheless, the number of parameters (Params) will continue to grow, thereby increasing the model's complexity and potentially impeding training.

**Table 3.** The impact of quantity $n$ on network performance.

| n Number | mAP (%) | Params (M) |
|:---:|:---:|:---:|
| 0 | 84.32 | 73.24 |
| 1 | 87.24 | 74.38 |
| 2 | 88.46 | 78.42 |
| 4 | **90.41** | 79.57 |
| 8 | 88.66 | 96.52 |

### 4.4.3. Evaluate GAE

To verify the effectiveness of GAE, we investigated the improvement in parameter $a$ on model performance on the HRSC-2016 dataset.

From Table 4, it can be seen that selecting the gradient adaptive equalization loss function has a higher mAP than the baseline model with a loss function of smooth L1. When $a = 0.5$, we improved by 1.02% compared to the baseline model, and as the value of $a$ decreased, mAP gradually increased. When $a = 0.2$, the mAP with the best performance of 90.41% was achieved, and compared to the baseline mode, the mAP improved by 6.06%. This indicates that the loss function we designed balances the weights of high-quality samples during the training process, improving the training effect. But, as the value of $a$ continues to decrease, when $a = 0.1$, mAP actually decreases. This indicates that an $a$ value too small during the training will disrupt the balance weight of high-quality samples again.

**Table 4.** Impact of parameter a on model performance.

| Settings | mAP (%) |
|:---:|:---:|
| smooth-L1 | 84.35 |
| $a = 0.1$ | 87.85 |
| $a = 0.2$ | **90.41** |
| $a = 0.3$ | 88.64 |
| $a = 0.4$ | 88.59 |
| $a = 0.5$ | 85.37 |
| $a = 0.8$ | 84.02 |
| $a = 1.5$ | 79.82 |

### 4.5. Comparative Experiment

#### 4.5.1. Experiment on DOTA

Our method was compared with other recent methods on DOTA. As shown in Table 5, for some categories such as plane (PL), baseball diamond (BD), bridge (BR), ground track field (GTF), soccer ball field (SBF), roundabout (RA), and helicopter (HC), our detection results were the best, with AP values of 90.23%, 86.71%, 61.24%, 84.65%, 69.89%, 81.37%, and 71.48% respectively. By synthesizing 15 categories, the mAP of this paper reached 79.6%, which is the highest among all methods.

**Table 5.** Comparison of the mAP of different methods in the DOTA dataset and the AP of each category.

| Methods | PL | BD | BR | GTF | SV | LV | SH | TC | BC | ST | SBF | RA | HA | SP | HC | mAP |
|---|---|---|---|---|---|---|---|---|---|---|---|---|---|---|---|---|
| Single-stage: | | | | | | | | | | | | | | | | |
| DRN [4] | 88.91 | 80.22 | 43.52 | 63.35 | 73.48 | 70.69 | 84.94 | 90.14 | 83.85 | 84.11 | 50.12 | 58.41 | 67.62 | 68.60 | 52.50 | 70.70 |
| R3Det [29] | 88.76 | 83.09 | 50.91 | 67.27 | 76.23 | 80.39 | 86.72 | 90.78 | 84.68 | 83.24 | 61.98 | 61.35 | 66.91 | 70.63 | 53.94 | 73.79 |
| FFA3 [30] | 88.80 | 74.40 | 48.90 | 57.90 | 63.60 | 75.90 | 79.60 | 90.80 | 80.30 | 82.90 | 54.30 | 60.00 | 66.90 | 66.80 | 42.50 | 68.90 |
| GGHL [31] | 89.74 | 85.63 | 44.50 | 77.48 | 76.72 | 80.45 | 86.16 | 90.83 | 88.18 | 86.25 | 67.07 | 69.40 | 73.38 | 68.45 | 70.15 | 76.95 |
| DAL [6] | 88.68 | 76.55 | 45.08 | 66.80 | 67.00 | 76.76 | 79.74 | 90.84 | 79.54 | 78.45 | 57.71 | 62.27 | 69.05 | 73.14 | 60.11 | 71.44 |
| RIDet-O [32] | 88.94 | 78.45 | 46.87 | 72.63 | 77.63 | 80.68 | 88.18 | 90.55 | 81.33 | 83.61 | 64.85 | 63.72 | 73.09 | 73.13 | 56.87 | 74.70 |
| CADNet [33] | 87.80 | 82.40 | 49.40 | 73.50 | 71.10 | 63.50 | 76.60 | 90.9 | 79.20 | 73.30 | 48.40 | 60.90 | 62.00 | 67.00 | 62.20 | 69.90 |

**Table 5.** *Cont.*

| Methods | PL | BD | BR | GTF | SV | LV | SH | TC | BC | ST | SBF | RA | HA | SP | HC | mAP |
|---------|----|----|----|-----|----|----|----|----|----|----|-----|----|----|----|----|-----|
| Two-stage: | | | | | | | | | | | | | | | | |
| ICN [12] | 81.36 | 74.30 | 47.70 | 70.32 | 64.89 | 67.82 | 69.98 | 90.76 | 79.06 | 78.20 | 53.64 | 62.90 | 67.02 | 64.17 | 50.23 | 68.16 |
| RRPN [34] | 88.52 | 71.20 | 31.66 | 59.30 | 51.85 | 56.19 | 57.25 | 90.81 | 72.84 | 67.38 | 56.69 | 52.84 | 53.08 | 51.94 | 53.58 | 61.01 |
| SCRDet [1] | 89.98 | 80.65 | 52.09 | 68.36 | 68.36 | 60.32 | 72.41 | 90.52 | 87.94 | 86.86 | 65.02 | 66.68 | 66.25 | 68.24 | 65.21 | 72.61 |
| $A^2$RMNet [35] | 89.84 | 83.39 | 60.06 | 73.46 | 79.25 | 73.07 | 87.88 | 90.90 | 87.02 | 87.35 | 60.74 | 69.05 | 79.88 | 79.74 | 65.17 | 78.45 |
| FAOD [36] | 90.21 | 79.58 | 45.49 | 76.41 | 73.18 | 68.27 | 79.56 | 90.83 | 83.40 | 84.68 | 53.40 | 65.42 | 74.17 | 69.69 | 64.86 | 73.28 |
| FR-Est [37] | 89.63 | 81.17 | 50.44 | 70.19 | 73.52 | 77.98 | 86.44 | 90.82 | 84.13 | 83.56 | 60.64 | 66.59 | 70.59 | 66.72 | 60.55 | 74.20 |
| CenterMap [38] | 88.88 | 81.24 | 53.15 | 60.65 | 78.62 | 66.55 | 78.10 | 88.83 | 77.80 | 83.61 | 49.36 | 66.19 | 72.10 | 72.36 | 58.70 | 71.74 |
| **Ours** | **90.23** | **86.71** | **61.24** | **84.65** | 70.09 | 80.54 | 88.01 | 90.45 | 87.46 | 76.89 | **69.89** | **81.37** | 78.91 | 68.89 | **71.48** | **79.38** |
| Each abbreviation is represented as: | | | | | | | | | | | | | | | | |
| Full Name | plane | baseball diamond | bridge | ground track field | small vehicle | large vehicle | ship | tennis court | basket-ball court | storage tank | soccer-ball field | round-about | harbor | swim-ming pool | heli-copter | – |

This method aligns the rotation convolution with the target, enabling more accurate detection of targets with rotation angles. These results indicate that the key to achieving precise detection of aerial targets is to construct an adaptive feature extractor for rotating targets and design feature filters that can characterize the direction of the target. At the same time, in the training set stage of the network, the model training efficiency is improved by optimizing the gradient imbalance problem. These two modules work together to ensure maximum improvement in network performance.

Figure 5 shows the detection performance of this method on the DOAT dataset, with the first row of images containing a large number of small targets of different categories and densely arranged. Our method can clearly detect every target, which proves that the dynamic rotation convolution proposed has good adaptability to direction, making the extracted features closely fit small targets, thereby improving the network's detection ability for small targets. The ships, docks, and other targets in the second row of images are distributed in multiple directions, and this method can also accurately detect each target direction. This suggests that the DCM module employed in this study enhances the network's ability to model target directional features, enabling the accurate detection of multiple targets with distinct directional disparities.

### 4.5.2. Experiment on HRSC-2016

We compared this method with other methods on the HRC-2016 dataset. From Table 6, it can be seen that compared to other methods, our method achieves 1.11% and 0.81% higher AP values than $R^3$Det and AR2Det and is superior to the other one-stage and two-stage detectors in the table. For targets with high aspect ratios and significant scale differences in aerial targets, the detection performance of this method is better.

Figure 6 shows the detection results of our method on the HRSC-2016 dataset. The first row of images contains a large number of targets with significant scale differences. Our method can accurately detect ships of various scales. The ship in the second row of images has the characteristics of a high aspect ratio and a dense multi-directional layout, and the detection box can better fit the target. This indicates that DCM optimizes the convolution feature extraction method and is more suitable for multi-directional aerial targets.

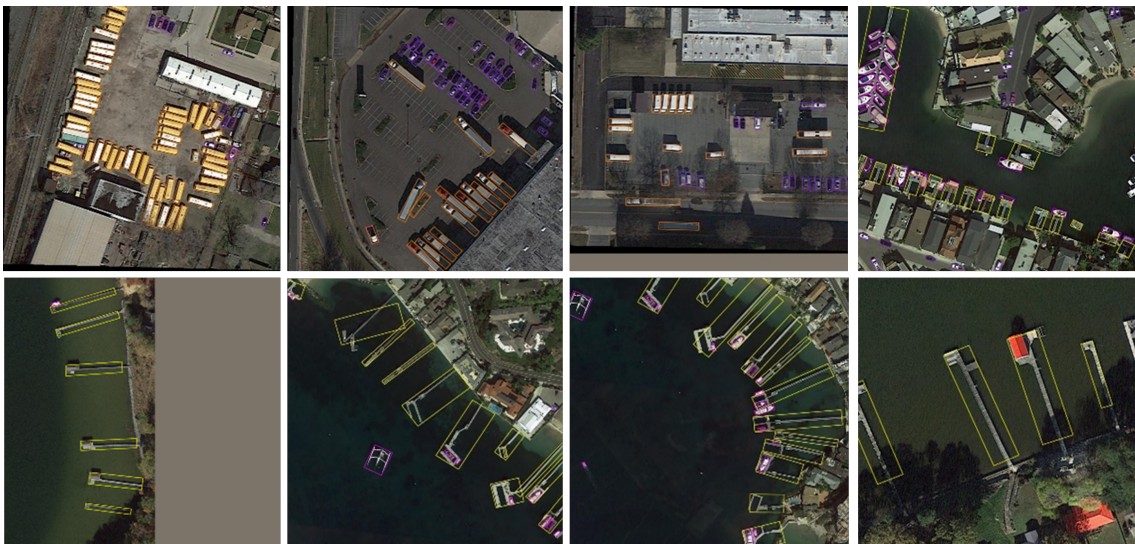

**Figure 5.** Visualize the results of this method on DOTA.

**Table 6.** Comparison of different single-stage and two-stage methods on the HRSC-2016 dataset.

| Methods | Backbone | Size | mAP (%) |
|---|---|---|---|
| Two-stage: | | | |
| $R^2$CNN [39] | Res101 | $800 \times 800$ | 73.1 |
| $R^2$PN [40] | VGG16 | - | 79.6 |
| AOPG [41] | Res50 | $800 \times 800$ | 80.6 |
| RoI-Trans [2] | Res101 | $512 \times 800$ | 86.2 |
| Single-stage: | | | |
| $R^3$Det [1] | Res101 | $800 \times 800$ | 89.3 |
| RRD [42] | VGG16 | $384 \times 384$ | 84.3 |
| OPLD [43] | Res101 | $800 \times 800$ | 88.4 |
| AR2Det [44] | Res101 | $512 \times 512$ | 89.6 |
| SDet [45] | Res101 | $800 \times 800$ | 89.2 |
| **ours** | Res101 | $800 \times 800$ | **90.41** |

### 4.5.3. Experiment on UCAS-AOD

As shown in Table 7, comparing different methods on the UCAS-AOD dataset, our method was the best, with an mAP of 90.52%.

**Table 7.** Comparison with multiple methods on UCAS-AOD.

| Methods | Car | Airplane | mAP (%) |
|---|---|---|---|
| R-RetinaNet [46] | 84.65 | 85.46 | 78.19 |
| R2PN [40] | 76.74 | 88.66 | 78.63 |
| RoI-Trans [2] | 88.02 | 90.02 | 89.02 |
| S2ANet [47] | 89.56 | 90.42 | 89.99 |
| RIDet-O [32] | 88.88 | 90.35 | 89.62 |
| ours | 86.64 | 94.28 | 90.52 |

Figure 7 shows the detection results of this method on the UCAS-AOD dataset. In the first row of images, the vehicle is obstructed by trees and buildings. This network is sensitive to the features of the target spatial direction, so it can detect partially occluded targets by predicting the direction and width of vehicles. The airplane in the second row of images exhibits characteristics with different directions and a dense layout. We ensured the accuracy of model detection through balanced loss training.

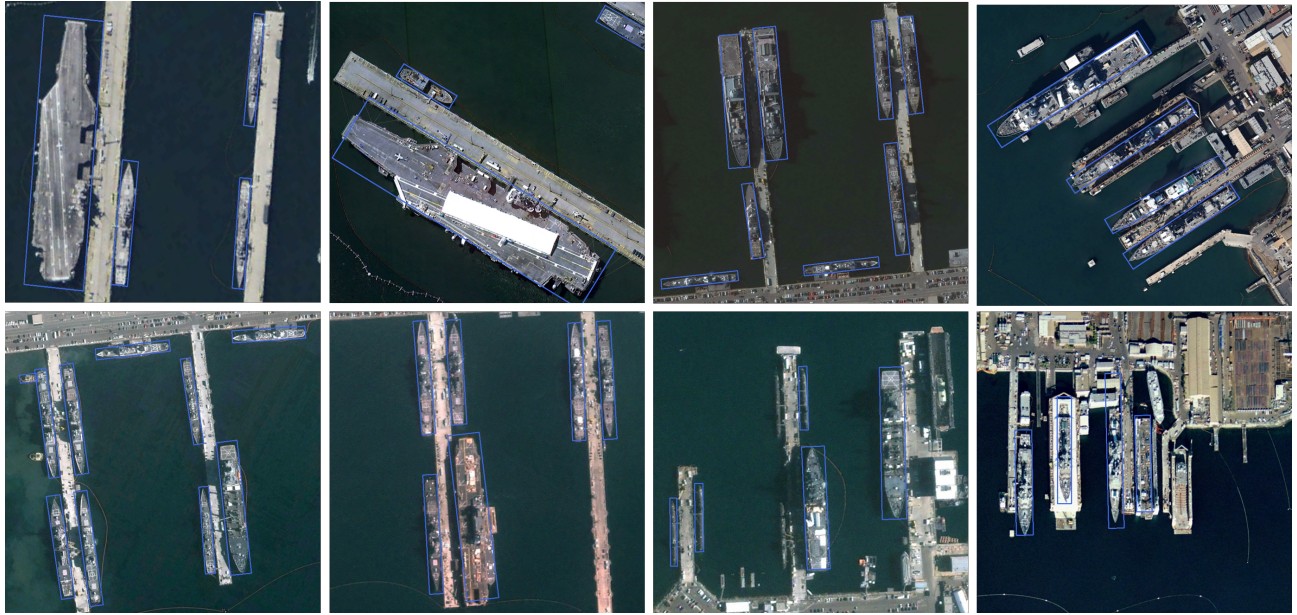

**Figure 6.** The detection effect of this method on HRSC-2016.

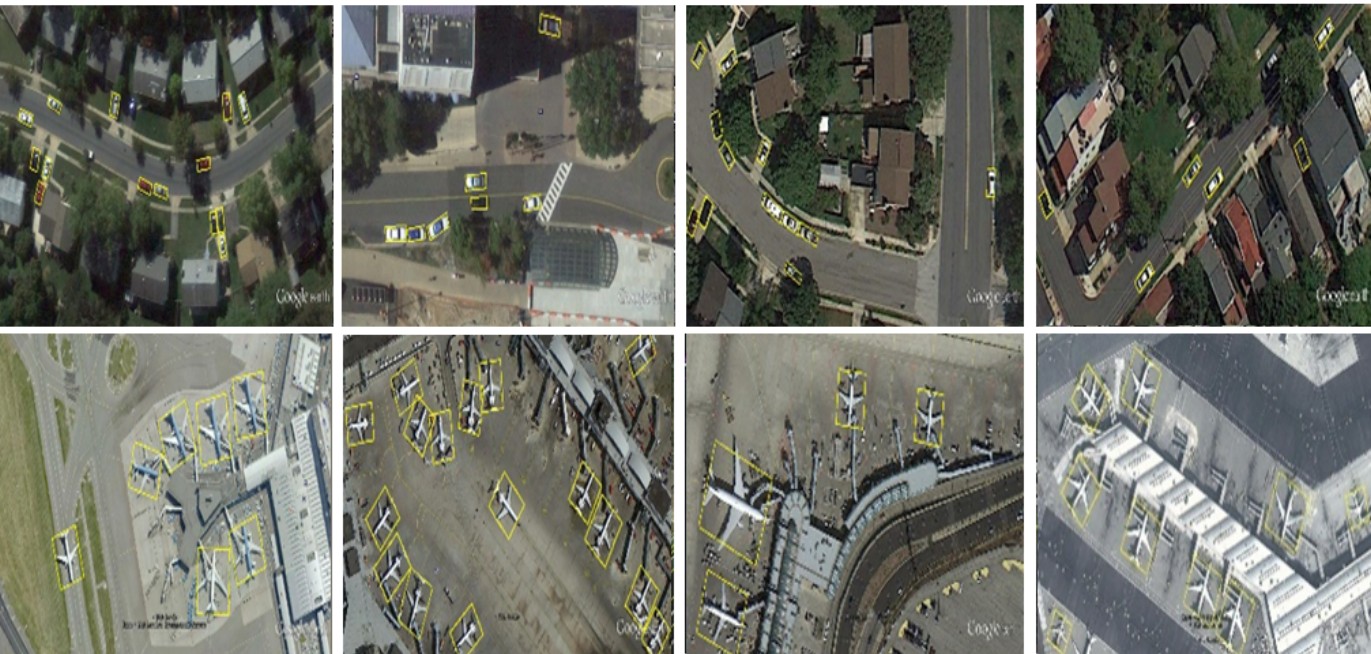

**Figure 7.** The detection effect of this method on the UCAS-AOD. The yellow box represents detection results.

Table 8 compares the inference time and mAP values of different models. Our proposed method has a significant improvement in model size compared to Faster RCNN and SSD, with significantly reduced Params and FLOPs of 133.9M and 65.1G, respectively. Running on 11th Gen Intel (R) Core (TM) i7 with hardware conditions, our fps reached a

maximum value of 8 and mAP also reached a maximum of 86.8%. Table 8 demonstrates that our method has improved accuracy and inference time compared to other models.

**Table 8.** Comparing the inference time of different models.

| Methods | Params (M) | FLOPs (G) | fps | mAP (%) |
| --- | --- | --- | --- | --- |
| Faster RCNN [48] | 361.1 | 100.5 | 1 | 83.2 |
| SSD [8] | 190.6 | 212.6 | 3 | 81.7 |
| CenterNet [49] | 124.0 | 62.3 | 4 | 84.3 |
| ours | 133.9 | 65.1 | 8 | 86.8 |

## 5. Conclusions

We systematically analyzed the problem of feature misalignment and imbalanced sample training in the current model for RS detection. To begin, in the backbone network, we devised an adaptive rotation convolution that aligns with the spatial direction of the target. The convolution kernels dynamically rotate based on the orientation of the aerial target. Different branches were designed to enable the network to efficiently capture the directional features of multiple targets in the image. The designed DCM module can be plug-and-play ported to any backbone network with convolution layers. Secondly, an adaptive equalization loss function was designed during the training to improve the contribution of high-quality samples and to ensure the stability of sample training. The effectiveness of the method has been proven on three common datasets: DOTA, HRSC-2016, and UCAS-AOD.

**Author Contributions:** Conceptualization, L.Y. and D.J.; Methodology, L.Y.; Software, Y.Z.; Validation, D.J. and Y.Z.; Formal analysis, D.J.; Investigation, X.L.; Resources, S.H.; Data curation, Y.Z.; Writing—original draft preparation, L.Y.; Writing—review and editing, D.J.; Visualization, L.Y.; Supervision, D.J.; Project administration, L.Y.; Funding acquisition, D.J. All authors have read and agreed to the published version of the manuscript.

**Funding:** This research received no external funding.

**Informed Consent Statement:** Not applicable.

**Data Availability Statement:** The HRSC-2016, DOTA, and UCAS-AOD datasets are available at following sites: https://aistudio.baidu.com/datasetdetail/54106, https://captain-whu.github.io/DOTA/dataset.html, and https://github.com/Lbx2020/UCAS-AOD-dataset, accessed on 6 January 2024, respectively.

**Conflicts of Interest:** The authors declare no conflicts of interest.

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
