# Peer review of "A Rotating Object Detector with Convolutional Dynamic Adaptive Matching"

_applsci, doi:10.3390/app14020633_

Round 1

Reviewer 1 Report

Comments and Suggestions for Authors

find the attached file 

Reviewer 2 Report

Comments and Suggestions for Authors

The received article "A Rotating Object Detector with Convolutional Dynamic Adaptive Matching" presents a new structure to improve the performance of CNN networks to effectively extract high aspect ratio and multi-directional features of aerial targets. In general, the manuscript has examined different aspects of the problem with different experiments. Observing the following points will help to make the article richer:

Comment1:

The caption of the figures does not fully explain the figures. For example, the caption of Figure 2 is insufficient.

Comment2:

It is better to provide the access link of each dataset in the dataset section.

Comment3:

The conclusion section is presented very briefly. more points need to be mentioned in it.

Comment4:

One of the important parameters in choosing a model is its execution time. Examining the execution time of the proposed model with other models can reveal the advantages or disadvantages of the proposed method.

Reviewer 3 Report

Comments and Suggestions for Authors

I am not at all convinced about the novelty of the mechanism. This is a very well addressed topic but the authors have done a poor job for the literature review. Take for instance this paper:

Hua, Z. et al., "AF-OSD: An Anchor-Free Oriented Ship Detector Based on Multi-Scale Dense-Point Rotation Gaussian Heatmap," remote sensing, 2023, 15, 1120. https://doi.org/10.3390/rs15041120

Some of the concepts are kind of similar to what is presented in the above paper. As for the amount of contribution, I would opine that the work may be better suitable for a conference. The technical novelty is low and the key parts have similarity with already existing ideas and concepts. Hence, I am not in favor or this work.

Comments on the Quality of English Language

Okay, but may need more polishing.

Reviewer 4 Report

Comments and Suggestions for Authors

After reading the paper entitled: "A Rotating Object Detector with Convolutional Dynamic Adaptive Matching", I have to say that the paper is recommended for possible publication after major revision. The following issues that need to be considered or answered.

1- Explain in details the contents of the equations, for example, equation (2).

2-The figure 4 is not clear. What is explain? 

Round 2

Reviewer 1 Report

Comments and Suggestions for Authors

no comments

Reviewer 2 Report

Comments and Suggestions for Authors

Accept

Reviewer 3 Report

Comments and Suggestions for Authors

Not convinced. Strong reject.

Comments on the Quality of English Language

Must be improved.
